# Optimisation of Embodied Carbon and Compressive Strength in Low Carbon Concrete

**DOI:** 10.3390/ma15238673

**Published:** 2022-12-05

**Authors:** Promise D. Nukah, Samuel J. Abbey, Colin A. Booth, Ghassan Nounu

**Affiliations:** 1School of Engineering, College of Arts, Technology and Environment, University of the West of England, Bristol BS16 1QY, UK; 2Centre for Architecture and Built Environment Research (CABER), College of Arts, Technology and Environment, University of the West of England, Bristol BS16 1QY, UK

**Keywords:** embodied carbon, optimisation, low carbon concrete, water-to-binder ratio compressive strength, simplex algorithm

## Abstract

To improve the prediction of compressive strength and embodied carbon of low carbon concrete using a program algorithm developed in MATLAB, 84 datasets of concrete mix raw materials were used. The influence of water, silica fume and ground granular base slag was found to have a significant impact on the extent of low carbon concrete behaviour in terms of compressive strength and embodied carbon. While the concrete compressive strength for normal concrete increases with reducing water content, it is observed that the low carbon concrete using lightweight aggregate material increases in compressive strength with an increase in embodied carbon. From the result of the analysis, a function was developed that was able to predict the associated embodied carbon of a concrete mix for a given water-to-cement ratio. The use of an alkaline solution is observed to increase the compressive strength of low carbon concrete when used in combination with ground granular base slag and silica fume. It is further shown that ground granular base slag contributes significantly to an increase in the compressive strength of Low carbon concrete when compared with pulverised fly ash. The optimised mix design program resulted in a 26% reduction in embodied carbon and an R^2^ value of 0.9 between the measured compressive strength and the optimised compressive strength.

## 1. Introduction

Embodied carbon encompasses carbon dioxide associated with extracting, manufacturing, transporting, and installing building materials. With an average of 66 billion square feet of buildings in construction every year, there are about 3.8 billion metric tons of CO_2_ emitted per year. The Intergovernmental panel on climate change (IPCC) projected zero-carbon emissions by 2050 if a 2 °C drop in global temperature is to be maintained [1]. With an ever-increasing global population, the need for more buildings is eminent with the tendency of increasing embodied carbon associated with building construction. Policy action to mitigate the effect of embodied carbon has resulted in the inclusion of whole life cycle analysis of buildings and tracking the building product environment impacts as part of a toolkit to enhance best practices for building sustainability efficiency. To reduce embodied carbon in concrete, the reduction and replacement of some concrete materials is a necessary strategy. Most industrial waste materials have been shown to be good replacements for coarse aggregate to produce lightweight aggregate concrete which enhances sustainability impacts on the environment while the replacement for cement in concrete is still a subject of many studies. While consideration is also given to optimising concrete and assessing its impact on carbon emission, the use of numerical solutions such as Artificial Neural Networks (ANN) has gained prominence in recent times. Notwithstanding, the effectiveness of ANN is predicated on a pool of efficient data to simulate the desired outcome. This makes it difficult to use ANN in analysing and effectively predicting a concrete dataset for which low carbon materials variables are scarce and lacking. This makes the use of the simplex algorithm a necessary tool for the optimisation of a small-size dataset as proposed in this study.

Concrete as the most frequently used material after water uses cement as its major raw material and has been noted to be responsible for a very high carbon footprint [2]. Carbon dioxide emission is associated with environmental damage, depletion of natural life and causing climate change. Carbon dioxide, as an active member of the greenhouse gas (GHG) family, is responsible for damage to the environment in the form of global warming and climatic change [3]. Cutting down on cement dosing in concrete requires the effort of both the policy makers as well as re-engineering the availability and accessibility of raw materials for low carbon concrete (LCC).

It has been shown that the built environment contributes about 10% of the global annual emission while in 2020, it accounted for 37% of the global emission [2]. The application of safety cautions due to uncertainties during design has contributed to most of the carbon footprint in concrete with carbon dioxide seen to be prevalent in cement used as raw materials for the construction of roads, bridges, and buildings, etc. [4]. The replacement of cement in concrete has been the target of much research over the years with concern on cement replacement using suitable supplementary cementitious material (SCM). Due to the potency of cement to enhance structural stability, a complete replacement is still a challenge to many studies. For instance, the use of SCM with good cementitious properties poses the challenges of low heat of hydration to produce the desired strength at an early age. The creation of this and other challenges add to the complexity of a complete replacement of cement. The use of chemical and physical properties of concrete raw materials have been subjected to laboratory evaluation to understand the concrete behaviour; however, due to the dynamics of concrete, the prediction of its behaviour for low carbon value at high compression is still a subject of speculation with very limited and scarce dataset availability. The introduction of numerical solutions has brought relief to most of the questions and limitations to researchers in that regard. One of the innovative solutions is the introduction of an Artificial Neural Network (ANN) as a predictive mechanism that can simulate the behaviours of concrete at different ages as well as different mix proportions. The effectiveness of ANN convergence is found to be dependent on the size of the dataset, the number of hidden layer neurons and the size of the learning rate parameters. The solution is obtained by trial and error since there is no defined and structure algorithm.

The application of ANN in the effective prediction and simulation of concrete behaviour is shown to be effective when there is a large repository of an effective dataset, which will ascertain the condition for which the validity of the prediction can only be ascertained and accepted. As the production of low carbon concrete is limited to the scarcity of dataset due to the variability of sustainable alternative concrete binders, the effectiveness of the ANN model for a small and limited dataset becomes a concern. A simplex algorithm as a linear program that uses the optimality of a decision to predict desired outcomes, which is engineered with the ability to optimise small-scale data for effective prediction and decision-making is used in this study.

## 2. Literature Review

Due to the cost intensity of the experimental program and uncertainty in the desired outcome, the use of numerical solutions to simulate and forecast likely outcomes cuts down on cost and time. The sensitivity of reinforced concrete pipeline deflection to the different soil filled depths was optimised by [5] using a finite element analysis program to check for the extent of shear reinforcement required. The use of MATLAB to optimise concrete compressive using a nature-inspired algorithm has been demonstrated by [6] and was shown to improve concrete compressive strength from 20 MPa to 90 MPa. However, there was an increase in carbon emission from 360 kgco_2e_ to 500 kgco_2e_ for 1 m^3^ of concrete. Similarly [7] studied the optimisation of post-tension bridges with the aim of reducing carbon emission by reducing the bridge deck depth using different slenderness ratios. The results show that a cost increase of less than 1% corresponds to more than 2% increase in carbon emission. The influence of machine learning to predict the effect of shear resistance on slender reinforced concrete beams without considering shear reinforcement was investigated by [8] and was concluded to have good performance compared to other models. Another application of the reduction in embodied carbon in concrete retaining structures was reported by [9] for which an increase in the reinforcement was noted due to reduced carbon emission.

Predicting concrete behaviour using ANN has been conducted by [10,11] on a dataset from 340 concrete mixes. Several methods of concrete mixing for high strength have been proposed by [12,13] using ANN. The use of MATLAB has attracted the attention of researchers in providing engineering solutions for both concrete and its raw materials. The cement hydration behaviour model called HYDCEM, written in MATLAB was developed by [14] using oxide composition, cement phase densities, species molar mass, phase and product densities and heat of hydration enthalpies as input data. A MATLAB program developed by Zealakshmi, Ravichandran and Kothandaraman [15] resulted in optimised high-strength concrete of 60 MPa. However, the use of MATLAB to reduce the embodied carbon in concrete while optimising the compressive strength is still scare and limited. 

The overdosing of cement in concrete is mostly responsible for the high carbon that is experienced in concrete. Construction materials use concrete and steel reinforcement due to their age-long benefits with cement and aggregate being the main raw materials of concrete which is noted for a very high carbon (CO_2_) value [16]. The production of cement involves the heating a mixture of limestone, shells, and chalk or marl combined with shale, slate, blast furnace slag, silica sand, and iron ore using fossil fuels to burn them at a temperature of more than 1400 °C in a kiln. Carbon is, therefore, released during cement production: roughly 40% of the Carbon generated is from the burning of fossil fuels in the manufacturing process and about 60% is generated during processing from naturally occurring chemical reactions. Consequently, an average of 927 kg of Carbon are emitted for every 1000 kg of Portland cement. Steel reinforcement contains a greater percentage of carbon to the extent that the strength of steel rebar is dependent on its carbon content. Cement is a major constituent of reinforced concrete, which is an integral part of the built environment; it has been identified to contribute to global carbon emission by about 6% [1]. A combination of concrete and reinforcement is responsible for 65–70% of the total embodied carbon in a building [17]. The reduction in embodied carbon in concrete requires the development of low carbon concrete (LCC) with consideration for sustainable replacement materials for cement. The use of lightweight aggregate as a replacement for normal-weight aggregate portends the benefit of sustainability since lightweight aggregate is often made from recycled aggregate from industrial waste.

The development of LCC using lightweight aggregate requires the production of concrete with raw materials of low carbon footprint with the potential for better performance in terms of structural integrity and sustainability. In addition, the use of lightweight aggregate offers the benefits of reductions in the self-weight of the structure which eventually reduces the dead load transfer to the foundation. This reduction in self-weight offers an advantage for a clear span beam construction compared to conventional concrete where a high self-weight poses a threat in addition to the external load the structure needs to resist [18]. The ability of a lightweight aggregate to improve thermal and noise insulation has been studied by Mastali et al. [19] and Real et al. [20]. It was shown that lightweight aggregate concrete exhibited a lower elastic modulus which is proven to enhance the better seismic performance of a structure by elongating the period of natural vibration associated with good deformation [21]. Notwithstanding, the issue of increasing cement content to maintain the required compressive strength still poses a threat to maintaining the low carbon value as well as its structural integrity. It is for this reason, that the optimisation of the materials constituting lightweight concrete is pertinent.

Kanavaris et al. [22] use lightweight aggregate concrete (LWAC) consisting of lytag as a coarse aggregate with a cement replacement of 40–60% cement (CEM I) using ground granulated blast-furnace slag (GGBS). The cement in the concrete was reduced to 40% without any remarkable distortion of workability, pumpability, strength and density of the concrete; however, it was observed that an increase in limestone powder resulted in lower workability. From the studies, it was shown that a higher proportion of cement results in higher values of embodied carbon. The overall embodied carbon reduction was about 6–7% owing to cement replacement with GGBS from 40 to 60%. It is worth mentioning that the use of more than five concrete raw materials as variables is necessary to determine the viability of the concrete in terms of compressive strength due to the dynamic response of concrete to different constraints. In this study, 12 concrete raw materials were used as design variables to optimise the compressive strength with its associated embodied carbon to obtain a low carbon high concrete (LCHC). With the consciousness for reduced embodied carbon becoming apparent, determining the benchmark for certification in view of the need to reduce embodied carbon in concrete, the optimisation of the concrete trial mix batch will help to reduce material wastage and time. Compared to an ANN where a large repository of a dataset is required for the effectiveness of prediction of the network in terms of training, validation and testing, the use of the simplex algorithm as demonstrated in this study aids the viable and deterministic prediction of concrete compressive strength with associated embodied carbon using a limited pool of datasets. While the expected output of the ANN changes for each epoch, it poses the difficulty of benchmarking a changing input variable to a constant expected output; the algorithm presented in this study allows for the manipulation of the input variables at a constant expected output. With the derived function, the extent of the expected embodied carbon can be predetermined using the effective water to binder ratio. The greatest challenge with concrete is finding the ability to model and simulate its performance in practice. The prediction of concrete is conducted by looking at its raw material characteristics, ambient temperature, etc., using algorithms that respond to variables such as changes in weather conditions. The idea of using the simplex algorithm in this paper is to reduce to dosage of cement in concrete using a small available dataset.

The article is organised into sections. Section A reviews the most recent literature on the method of optimisation, research trends and the result obtained. Section B focuses on low carbon concrete and embodied carbon concrete dependent variables. Section C discusses the process of formation of the algorithm for this review. Finally, section D analyzes the result and presents the extent of optimisation of embodied water in the concrete mix.

### 2.1. Use of Nanomaterials in Concrete 

The use of nanomaterials has received increasing interest in reducing concrete greenhouse gas emissions. During cement hydration, the structure of the concrete materials condensed in size by nanometres using the calcium–silicate–hydrate (C–S–H) gel. This is because the particle size of cement ranges between 7 and 200 nm while that of aggregate has a particle size from millimetres to centimetres. It is then obvious that concrete can be described as a nanomaterial.

The particle sizing of these materials ranges from 1 to 100 nm. The effectiveness of concrete carbon sequestration is enhanced with effective carbon hydroxide absorption and the reduction in chloride ion migration along with its diffusion coefficient. Studies have shown that 2% of nano-SiO_2_ reduces concrete water absorption by 58% while 3% nano-SiO_2_ reduces water absorption by 36.84%. This reduction in water absorption resulted in an improvement in concrete mechanical properties by densifying the concrete matrix during the hydration reaction. Available nanomaterials include Carbon nanotubes (CNTs), nano-SiO_2_ (nS), nano-Al_2_O_3_ (NA), graphene oxide (GO), nano-TiO_2_ (NT), nano clay (NC), nano-ZnO_2_ (NZ), and nano-Fe_2_O_3_ (NF) [23,24].

Regarding the effect of concrete on embodied carbon, Tabrizikahou and Nowotarski [25] demonstrated that concrete and steel are major contributors to embodied carbon from primary building materials to up to 55% of the total materials needed. The study was carried out using CEMI, nano silica, fly ash, and metakaolin at clay incineration to 800 °C as cement material optimisation. Optimisation in the design of concrete structures has been studied by Yeo and Gabbai [26] and it was reported that 10% of the building's embodied carbon was reduced using the program algorithms with an objective function that minimises embodied carbon in concrete design, subject to some design parameters. The design parameters include Factored moment Mu, Factored shear force, Concrete compressive strength, longitudinal reinforcement, etc. Similar literature has shown that embodied carbon can be reduced by adopting certain techniques which include the use of low carbon materials, material minimisation strategies, construction optimisation strategies, local sourcing of materials, material reuse, and recycling strategies [27,28].

Similarly [29,30] have shown the potency of reducing the embodied carbon of buildings by developing design alternatives through varying the structural forms of the building, floor system, reinforcing technique and layout.

As the building structure comprises mainly a floor slab and beam, the options of modelling the floor slab and beam using a thin shell concrete from finite elements were explored as a design concept in reducing embodied carbon. The self-weight of this structural member was reduced to 44% and embodied carbon in the members was reduced to 57% [30]. Similar studies include the use of structural member geometry refinement to influence the embodied carbon content of the structure [31].

The effect of material selection on the carbon footprint of structures has also been extensively studied [22,32,33,34,35]. Kim, Tae and Roh [32] deduced optimal mix design in assessing the emission of embodied concrete from the mix design database. It was reported that 7% of embodied carbon was reduced. Conservative concrete was developed using experimental test results for 13 concrete mixes for 50%, 60%, 70% and 80% GGBS replacement with Ordinary Portland cement (OPC) [33]. The study reported an embodied carbon reduction of 154 kg/m^3^ for 80% GGBS cement replacement which confirms that GGBS significantly reduced embodied carbon as an SCM binder material. Thilakarathna et al. [34] reported reducing embodied carbon using machine learning algorithms with consideration for nine concrete raw materials as input variables

Recent studies by González and García Navarro [36] have shown that there is a possibility for a decrease of about 30% in CO_2_ emissions when conventional materials were replaced with lower embodied carbon material, which makes the issue of materials replacement a major consideration in embodied carbon reduction. The benefits of the application of low carbon concrete in construction will include but are not limited to enhanced physical and mechanical properties, reduction in raw materials and efficient energy consumption [37,38]. The enhancement of concrete using sustainable pozzolanic materials as cement replacement will improve its ability to increase compressive strength and make it suitable for fire resistance, light and enhance high durability. There are different categories of high thermal insulation of concrete and in most cases, the optical properties are optimised using low carbon materials [39]. The property of construction material components with the ability to reduce the surface temperature of a building emanating from solar energy is referred to as its optical property [40]. Concrete, therefore, has been noted as a temperature storage material which eventually calls for the need for an efficient concrete energy-conservative material component [39,41,42]. 

The solar reflectance effect is often reported in the magnitude of 0 to 1. The fibre material thermal conductivity of some selected fibre raw materials is shown in Table 1. Normal concrete thermal conductivity is observed to be ~2.25 W m^−1^ K^−1^ [43] which is high compared to that of fibre materials as shown in Table 1.

### 2.2. Sustainable Low Carbon Concrete (LCC) Materials

Low carbon concrete is concrete produced with consideration for the carbon footprint of the constituent materials in terms of sustainability and structural performance. The readiness and availability of concrete constituent materials with enhanced ubiquity during sourcing while placing emphasis on its low carbon content makes it a sustainable low carbon concrete material. Investigation of the impact of the water–cement ratio in concrete on its mechanical properties was studied by Proske et al. [44]. The concrete compressive strength, workability and durability were examined. It was observed that when the cement in the concrete was reduced from 270 to 100 kg/m^3^, an embodied carbon reduction of 35% was reported. Additionally, fillers and additives were utilised gradually in substitution for the cement. Similarly, a further reduction to more than 60% was recorded with the use of granulated blast-furnace slag (GGBS). The result from these studies underscores the fact that there is a structural response to each degree of optimisation in consideration of the water-to-binder ratio. While a decrease in water-to-cement for conventional concrete is reported to enhance the compressive strength, for an LCC using recycled aggregates it has been observed that a reasonable amount of water is required to attain a high compressive strength for a lightweight recycled aggregate concrete [45].

The use of waste or residual materials that uses a small amount of energy for production has the potential to produce concrete with a low carbon footprint. Cement is often replaced with materials such as silica fume, fly ash as well as wood ash. The work of Tam et al. [46] did show that recycled concrete is vulnerable to poor concrete compressive strength; however, about 40% of the recycled aggregate can be re-used [47] in concrete construction. Recycled fly ash has been used as a substitute for conventional high carbon concrete using a mixture of lime and water. This enhanced sustainable low carbon concrete was reported to exhibit low embodied carbon. Fly ash is effective in the resistance to concrete alkali–silica resistance which is known to be a problem of normal conventional concrete. It was noted by [48] that increasing the percentage replacement of cement using fly ash reduces the compressive strength as well as the modulus of rupture of concrete. Another alternative binder replacement material is ground granular base slag (GGBS). This is formed from the production of iron as industrial waste. Its molten nature known as slag is tapped off from the blast furnace and cooled rapidly. It has cementitious properties when reacted with water, but the heat of hydration is usually low, thus the heat of hydration can be increased when placed in an alkaline environment. The granular nature of slag from blast furnaces has been transformed into a molten state when used in concrete. It is a by-product material that can be recycled which makes it a sustainable material. It has the advantage of low heat of hydration during production. The extent of cement replacement due to its pozzolanic tendency can be achieved between 60 and 80% [49].

It was observed that the use of recycled aggregate is effective when used in combination with fly ash as supplementary cementitious material (SCM) but was not suitable when combined with GGBS [49]. The reaction of cement and slag resulted in a slow rate of hydration which eventually resulted in low mechanical strength at an early age and high strength at a later age [50]. Combining the cement–slag mixture with silica fume can help to increase its heat of hydration and increase its mechanical properties [51,52]. The use of ternary cement (Cement, slag and silica fume) reduces the rate of water and gas penetration into the concrete which helps to improve the mechanical properties; however, a reduction in concrete durability was observed [53]. The structural behaviour of cement as a mixture with silica fume and GGBS has been studied by Bonavetti et al. [54]. From the study, the mechanical strength of the mix increases when about 5% of silica fume and up to 65% of granulated blast furnace slag were used.

Silica fume is obtained from the production of silicon. It exists as a fine powder used in a slurry form to produce high-performance concrete. During the condensation of silicon dioxide and ferrosilicon alloy, silica fume is a by-product having the nature of an ultrafine power. It has the advantage of low permeability which is suitable to increase the concrete structural performance [55]. The combination of silica and slag at 10% to 15% of the binder recorded an improved compressive strength at a constant slump [56]. Other sustainable materials of concern are aggregate obtained from concrete construction demolition known as recycled aggregate. Recycled aggregate concrete (RAC) is an environmentally friendly construction material that is also used for some structural applications [57,58,59,60]. However, the volume of the mix in terms of recycled aggregate composition should be determined to ensure that structural performance is within the required acceptable standard [61].

### 2.3. Classification of Low Carbon Concrete

From the Concrete Industry Sustainability Performance Report 2018 on performance data [62], the climatic change and energy action on carbon is measured using sustainable principles based on its performance in delivering the embodied carbon target to achieve the sector climatic change agreement. 

The binder constitutes the core of embodied carbon in concrete which can partly be replaced with secondary cementitious material. This means that the binder CEM1 can be partially replaced with GGBS, pulverised fly ash or limestone, etc., to produce a low carbon concrete in a proportion to attain good structural performance. Table 2 shows the carbon factor for some LCC constituent materials. 

Concrete carbon content is measured as the embodied carbon associated with the measured slump as a benchmark for rating the embodied carbon in concrete which varies from A to E. From Table 3 and Table 4, it is observed that without SCM cement replacement for a low carbon target, the structural concrete for the ground floor, superstructure and high strength concrete will fall within the rating E (red zone) in terms of carbon emission. With SCM replacement at 30% pulverised fly ash (PFA), and 50% GGBS, there is a potential for an increase in the grade of concrete to carbon rating C and D, respectively, in terms of carbon emission. However, in consideration of concrete compressive strength, for an LCC to attain a rating of A+, there will be the need for further concrete mix optimisation using SCM, admixtures and additives with high pozzolanic activities. 

With preference from design options using low recommended minimum cement content and the allowable cement types, the highest levels of clinker replacement will ultimately reduce carbon emission in concrete. It is possible to predict maturity and strength using a time-temperature relationship since concrete strength is a function of the time between casting, testing, and curing temperature.

The boundaries to established low carbon concrete have been set by the Low Carbon Concrete Group of the Green Construction Board (LCCG) [45] of the institution of civil engineers (ICE), UK. In the report on Low Carbon Roadmap, the low carbon concrete rating shown in Table 3 and Table 4 depicts the extent of tolerance for sustainable low carbon concrete.

Following the strength rating of LCC, the embodied carbon rating for the different classes was recommended as shown in Table 5. An assessment of the Table 4 and Table 5 indicates that for an LCC of rating A+ to be achieved, there should be a minimal emission of 0% and a maximum of 5% emission as a threshold which presupposes that the compressive strength will fall below C28/35. With the replacement of cement with SCM, the minimum expected LCC carbon rating for foundation blinding is within the A+ at low compressive strength. From Table 3, we can observe that the use of GGBS is proven to be advantageous over PFA in terms of carbon reduction in concrete. It is shown that at 50% GGBS replacement, the embodied carbon is reduced from 165 Co_2e_/m^3^ for cement to 95 kg Co_2e_/m^3^. This implies that an increase in GGBS replacement will further reduce the embodied carbon. 

It is expected that a limit of consideration in terms of LCC grade from the concrete design process to the construction stage would limit the extent of embodied carbon emission from the built environment. Given the tolerable value of 95–190 kg Co_2e_/m^3^ for concrete in foundations, which lies within grade B, a further optimisation is likely to have a grade A for the foundation while not falling short of the required mechanical performance.

When considering the choice of sustainable concrete, the effect of carbon content outweighs the influence of other parameters on concrete while not ignoring structural integrity. From Table 6, it is shown that 100% of clinker addition during cement production is due to waste residues from coal-fired power stations (PFA) and blast furnace slag. This resulted in about 50–60% of emissions from limestone (CaCO_3_) decomposition to calcium oxide (CaO) from the combustion of fossil fuels and electricity consumption [64]. The use of alkaline activated concrete (AAC) has been shown to be more environmentally sustainable than ordinary Portland concrete. A decrease in the concrete pore volume after 28 days was attributed to the reduction in Embodied carbon for the AAC [65]. Alkali aggregate reactions are harmful phenomena that occur within the concrete when the alkalinity of the pore solution of the concrete is sufficiently high and wet. The gel formed as a product of the reaction absorbs water and causes concrete deterioration followed by volume expansion and the development of a swelling pressure [66]. The relationship between concrete compressive strength, shrinkage, pore size, porosity, shrinkage, and permeability has been studied by Narayanan and Ramamurthy [67]. From the study, concrete shows intense failure from shrinkage and low compressive strength with increasing porosity. From the foregoing, it is pertinent to observe that the behaviour of concrete constituent materials and performance at some level of optimisation is dynamic which influences the concrete behaviour. Considering the available laboratory sample test results and performing some level of optimisation for a reduced embodied carbon at optimal compressive strength will reduce the loss of materials and time when carrying out a mix design. The optimisation model most frequently used is the linear programming model which has shown a significant impact on sustainability, and economic and government policy decisions [68]. MATLAB and CUDA environments have been used to implement an optimisation of about 5000 variables with 5000 constraints to attain a speed level of 5.5 for an intel Core i7 3.4 GHz and NVIDIA Quadro system. A revised simplex was carried out to enhance a basis update showing better performance of expected sample results [69]. 

### 2.4. Significance of Study

To produce low carbon concrete (LCC) from previous studies, mix compositions were made from CEMI and other SCM materials at different mix proportions. Consideration from the literature has always been given mainly to the replacement of CEMI with SCM materials without considering the impact of water content on the carbon value. However, the effect of water is being considered only in terms of its workability which is measured by the slump value. The results of previous studies have focused on the correlation between compressive strength and embodied carbon. However, from the literature it was observed that the mechanical properties in terms of compressive strength of the low carbon concrete produced were limited and scarce; hence the need to consider the possibility of assessing the effect of water in the mix composition of low carbon concrete. The current study has established that the water composition in a concrete mix significantly influences the embodied carbon of the concrete for lightweight aggregate concrete via a series of optimisation techniques. This is possible because water tends to increase the porosity of concrete thereby enhancing the development of carbonation depth because the lightweight aggregate is obtained from recycled aggregate with a low density and with a tendency for high water absorption [22,32,33,34]. The use of sustainable SCM materials for low carbon concrete is likely to increase the need for water in the concrete which eventually affects the workability and the concrete strength. To reduce this effect, high water reducing agent (HWRA) has been proposed and used but not with an effective result to attain the required workability. Thus, a concrete mix optimisation is proposed that will match the required mix composition to its compressive strength with low embodied carbon using optimal water content.

The correlation of concrete raw materials to compressive strength and embodied carbon from Thilakarathna et al. [34] is shown in Figure 1. The input variables include Coarse aggregate (CA), Fine aggregate (FA), Cement (C), Ground granulated blast furnace slag (GGBS), Fly ash (FA), Silica fume (SF), Superplasticizer, Water, W/B and Fine aggregate (S). 

Figure 1 implies that it takes about 800 kg/m^3^ of cement to attain the value of the compressive strength and embodied carbon equivalent of a combination of GGBS, PFA and Silica fume at less than 200 kg/m^3^. This implies that the use of GGBS, PFA and Silica fume as SCM for low carbon concrete compressive strength optimisation significantly influences the determination of its mechanical behaviour.

## 3. Methodology

### 3.1. Data Source

MATLAB uses syntax that is suited for both scientific and engineering programming in a matrix and vector form for input and outputs. The applicable code is easily editable, especially with multifunctional scripts that allows the user easy access and flexibility. With the aim of accommodating all feasible concrete raw materials, the range of variables was extended to 12 concrete batches for one iteration. Due to the scarce and limited dataset for sustainable concrete and as part of the novelty to use the developed algorithm on a limited size of dataset, the sensitivity of the input parameters was examined from literature reviews of previous studies.

Analysis of the concrete from [22,32,33,34,35] was evaluated and extrapolated to 12 input concrete raw material variables. The dataset obtained were test results from laboratory experiments and observed to significantly impact the compressive strength and embodied carbon. The age of concrete for all data used in the study is 28 days and is kept constant without consideration for age as an input variable.

Concrete mix composition with the same compressive strength is observed to differ in the associated embodied carbon emitted. This also benchmarked the range of values of compressive strength for which certain values of embodied carbon can be estimated. A study of the assessment of embodied carbon emission of alkali-activated concrete substituted fly ash with metakaolin was carried out on low embodied carbon concrete and compared with Portland cement concrete (PCC). Calcium hydroxide and silica fume were used for the different mixes as an activator (admixture). It was reported that Ordinary Portland Cement (OPC) contributed to about 80% of embodied carbon in concrete [35].

CEMI cements are produced from ordinary Portland cements (OPC) with a maximum of 5% additional material. OPC materials are usually 0–5% gypsum and about 95–100% clinker. Cement has been identified as the component responsible for the high carbon value of concrete. This is mainly possible due to the materials required for its production which includes the catenation of limestone. A look at the embodied carbon value for concrete components in Table 7 shows that cement holds the highest embodied carbon factor in the concrete mix compared to other binders which makes the use of SCM a necessary consideration for a low carbon concrete. The limit of the input variables as shown in Table 8 allows for increment at each optimisation process.

The problem formulation is achieved by having an idea of the desired embodied carbon and compressive strength from the measured experimental value while being bounded by the design constraints, which are limited by the contribution of the proportion of the concrete variables and the measured outputs in terms of compressive strength and its associated embodied carbon. The measured compressive strength equates to a function defined by a proportion of the concrete variables, and it is repeated as a constraint for other measured output and represented as a matrix. An operation is performed to determine the entering and leaving variables using the non-negativity conditions expressed in Section 3.2. A collation of all matrices represents the simplex algorithm Tableau shown in Table 9, Table 10, Table 11, Table 12, Table 13, Table 14 and Table 15 and a solution for which there is no entering or leaving variable where the solution will be obtained at values corresponding to the optimal concrete mix proportion

### 3.2. Optimisation

Linear programming (LP) involves optimisation that meets both a linear objective function and a linear constraint. The laboratory results of low carbon concrete from [22,32,33,34,35] were selected and used. For an optimisation problem to be linear such that the simplex algorithm will apply, all constraints function as well as the objective must be linear to solve either a maximisation or a minimisation problem.

The standard linear programming problem can be written as 

Minimise/Maximise Y = d_1_ x_1_ + d_2_ x_2_ · · · + dn xn = ∑d_j_ x_j_

subject to the constraints

b_11 × 1_ + b_12 × 2_ + · · ·+ b_1n_x_n_ ≤ c_1_

b_21 × 1_ + b_22 × 2_ +· · ·+ b_2n_x_n_ ≤ c_2_

...……………………… 

bm_1 × 1_ +bm_2 × 2_ +· · ·+ bmnxn ≤ cn

(or Bx ≤ c) 

and

x_1_ ≥ 0, x_2_ ≥ 0, . . . , xn ≥0 (or x ≥ 0). 

Here, Y = d_1_ x_1_ + · · · + dn xn is the objective function, Bx ≤ c the constraint and x ≥ 0 is the non-negativity constant [72].

The embodied carbon associated with the various concrete mix proportions from some studies was collated and shown in Table 8, Table 9, Table 10, Table 11, Table 12, Table 13 and Table 14. Establishing the constitutive relations between embodied carbon and its concrete materials constituent is apparently scarce due to the variability in the data associated with the available literature [73]. 

Optimisation of their concrete mix design for a target minimum embodied carbon was conducted using a linear programming (LP) algorithm and implemented in a MATLAB environment. The program algorithm resulted in the creation of the script ‘m-file’ that is transformed into MATLAB code containing the functions as well as the scripts.

The objective function is to minimise the embodied carbon in one instance and maximise the concrete compressive strength at the second instance on data as reported in this literature, subject to variables in the concrete mix as shown in Table 9, Table 10, Table 11, Table 12, Table 13, Table 14 and Table 15. The value in the objective function is obtained from the output of embodied carbon and concrete compressive strength as reported in the literature. These two scenarios were considered in this paper. In implementing the maximisation problem, the problem statement of the objective function is established by taking the negative of the variable in the objective function that has impacted on minimising embodied carbon. The programming statement used in this study was such that a function f (an objective function) represents the concrete compressive strength and embodied carbon defined by the independent variable x (representing 12 concrete mix components) and the matrix vector c as a function of the independent variable. It is important to state that a simplex algorithm program was adopted to optimise the set of independent variables for the desired objective function. The mathematical statement is written as shown below

Minimise , f=c x; for Embodied carbon 

$Maximize\;,\;f=c^{T}x$; for compressive strength

subject to

Ax+s=b for (1<x<n) where n is the number of variables; $f:R\overset{n}{\to }R$ is the objective function. The superscript T denotes transpose operation. X denotes variables that are components of vector x, $C,x\in R^{n}$. $S_{i}=\left(b_{i}-a_{i}^{T}x\right)\geq 0$ is the slack variable variables, i=1,2⋅⋯n is the linear constraint function where a is the gradient.

The decision variable x is implemented as follows

$x_{1}=f$(OPC, in mix 1,2,3,..12),

$x_{2}=f($ FA,in mix 1,2,3,..12),

$x_{3}=f$ (CA, in mix 1,2,3,..12),

$x_{4}=f$(,W, in mix 1,2,3,..12),

$x_{5}=f$ (CH.A in mix 1,2,3,..12),

$x_{6}=f$ (GGBS in mix 1,2,3,..12),

$x_{7}=f$(PFA in mix 1,2,3,..12),

$x_{8}=f($NAOH in mix 1,2,3,..12),

$x_{9}=f$ (,MEK in mix 1,2,3,..12),

$x_{10}=f$ (NaSi, in mix1,2,3,..12),

$x_{11}=f$ (SF, in mix 1,2,3,..12),

$x_{12}=f($,w/c in mix 1,2,3,..12),

Where OPC = Ordinary Portland Cement,FA = Fine aggregate,CA = Coarse aggregate, w = Water, CH.A = Chemical admixture, GGBS = Ground granular base slag, PFA = Pulverised fly ash, NAOH = Sodium hydroxide solid, MEK = Metakaolin, NaSi = Sodium silicate, SF = Silica fume, EC = Embodied carbon, Concrete compressive strength.

For the minimisation of embodied carbon, the objective function z is implemented as follows:
$$Z=\left(embodied\;carbon\;in\;OPC\right)x_{1}+\left(embodied\;carbon\;in\;FA\right)x_{2}\;+\left(embodied\;carbon\;in\;CA\right)x_{3}+\left(embodied\;carbon\;in\;W\right)x_{4}\;+\left(embodied\;carbon\;in\;CH.A\right)x_{5}+\left(embodied\;carbon\;in\;GGBS\right)x_{6}\;+\ldots .$$

For the maximisation of compressive strength, the objective function z is implemented as follows
$$Z=\left(Compressive\;strength\;\right)x_{1}+\left(Compressive\;strength\;\right)x_{2}\;+\left(Compressive\;strength\;\right)x_{3}+\left(Compressive\;strength\;\right)x_{4}\;+\left(Compressive\;strength\;\right)x_{5}+\left(Compressive\;strength\;\right)x_{6+\ldots }$$

The decision variables were subject to the minimum value of the parameters. The concrete mix design result as obtained from the literature were reproduced in a spreadsheet. A total of 84 datasets were used, and the 12 concrete mix design batches for each set of optimisations.

### 3.3. Embodied Carbon, Water Cement Ratio, GGBS, and Admixture Relationship

Secondary Cementitious Materials (SCM) are materials used to replace Portland cement in a concrete mix primarily to reduce embodied carbon. Among the many SCMs available, GGBS is mostly preferred mostly due to its low carbon value and its ability to attain higher strength with age. However, as a water absorbent material, its hydration depends mostly on the presence of an activator which is mostly achieved by increasing the temperature of curing to between 50 °C and 80 °C to shorten its curing time. Other SCM includes flyash, metakaolin and silica fume [74].

Similarly, as the strength of concrete is enhanced also by its aggregate sizing, GGBS increases concrete strength also by closing the concrete pore and acting as a binder material. Closing the concrete pores is further made unrealistic by the much water absorbent nature of GGBS which requires that much more are added, thereby making the concrete porosity wider, impacting the embodied carbon negatively. The porosity of the concrete then allows for the formation of carbon (carbonation) in the concrete, which is made possible by the aggregate sizing, water-to-binder ratio (w/c) and the SCM used. It is observed that while a carbonation depth of 15 mm is likely to be formed at w/c of 0.6 in 15 years, for w/c of 0.45, the carbonation will last 100 years before it attains 15 m [75].

Admixtures such as superplasticizers are, therefore, introduced into the concrete mix to decrease the widening of the concrete capillaries which helps in reducing the formation of carbon and eventually reducing the embodied carbon.

While several studies have shown that embodied carbon increases with an increase in concrete strength for concrete using recycled aggregate [76], etc., it further means that a reduction in embodied carbon of concrete will accompany a reduction in concrete strength. As the carbonation of concrete is enhanced by the capillaries in the concrete (porosity) which depend on the coarse aggregate size, water–cement ratio, the concrete binder material (admixture) as well as the SCM in the mix. To optimise the mix design based on these parameters, a balance must be established such that the same Water that reduces embodied should improve the mechanical properties of the concrete. Table 9, Table 10, Table 11, Table 12 and Table 13 show the concrete mix design composition and their corresponding concrete strength and embodied carbon. The minimum value is obtained from several trials in the linear program resulting in an optimised embodied carbon.

The mixed batch considered for the optimisation has a total of 12 trial concrete mixes with material composition for each trial mix as shown in Table 9 below. From Figure 2, it is observed that 45% of cement and 15% of water for a concrete mix, resulted in concrete embodied carbon of 174.8 kg-Co_2e_/m^3^ at a compressive strength of 34.5 MPa. This is equivalent to a water–cement ratio of 0.3.

The concrete mix materials assign the desired behaviour for a given concrete use. This is influenced by its chemical and physical properties which often attain a complex structure because of the formation of new compounds, which is made possible by the hydration reaction of the binder. As the concerns for more sustainable and eco-friendly concrete gains momentum, the choice of materials for low carbon concrete cannot be over-emphasised. A practical and realistic approach will be to minimise cost and time on trial mixes which often result in material wastage and manpower loss. It becomes necessary to use the available tools of the linear program as an effective predictive method to select suitable concrete material composition from existing low carbon concrete data set.

Similarly, Figure 3 shows that 14% of concrete embodied carbon is associated with 4% of its compressive strength for the given concrete mix optimisation, with the percentage material composition as shown in Figure 3 and Figure 4. A reduction in embodied carbon from 179.23 kg-Co_2e_/kgm^3^ to 77 kg-Co_2e_/kgm^3^ was recorded using a water reduction from 150 kg/m^3^ to 75 kg/m^3^ at a minimal concrete strength decrease from 11.90 MPa to 10.2 MPa for a cement content of 150 kg/m^3^. This minimal decrease may be likened to the use of lytag aggregate compared to others. There was a remarkable increase in embodied carbon to 119.49 kg-Co_2e_/kgm^3^ and an increase in concrete strength to 32.258 MPa when 300 kg/m^3^ of cement was used. From Figure 3, Figure 4, Figure 5, Figure 6, Figure 7 and Figure 8, about 16% of water and 28% of cement of the concrete mix reduces the low carbon concrete to grade A, at a compressive strength of about 30 MPa which is shown to be tolerable in consideration of use and sustainability

A similar effect of cement and water on embodied carbon and concrete strength was observed, but it was noted that PFA does not significantly affect the embodied carbon in concrete without water as a control. A summary of the optimized embodied carbon is shown in Table 16 with a reduction of 26% and an associated reduction in compressive strength in Table 17 to 24%. The reduction in compressive strength is equivalent to an R^2^ value of 0.94 between the measured and the optimized concrete compressive strength.

The sustainability performance of Alkaline Activated Concrete (AAC) as demonstrated in Figure 8 shows the influence of the water binder ratio on the embodied carbon and compressive strength. This can be explained by the concrete hydration process considering the chemical composition of the concrete constituent raw material. Table 20 shows that Pulverised fuel ash (PFA) has aluminium and silicon in high proportions but less in calcium oxide (CaO) while GGBS has more of CaO, a combination of both, making a good alternative cementitious binder.

The performance of the proposed model is shown in Figure 9 with an R^2^ value of 0.94 in comparison with the measured dataset. The average value of compressive strength for 12 concrete mixes for each batch was taken. The wide deviation shown in batch 1 is evident from the value of 650 kg/m^3^ and 1400 kg/m^3^ assumed), respectively, for fine and coarse aggregate as the dataset was lacking in these values. The relationship between the measured and the optimised can be expressed by the function
$y=28\cdot 93log_{e}^{x}-73\cdot 589$ where x is the measured compressive strength.

### 3.4. Low Carbon Concrete and Embodied Carbon Dependent Variables

Energy efficiency is a concern that can be addressed within the purview of design and sustainability with consideration for available resources to enhance the minimal or near-zero carbon footprint of concrete. The chemical composition of concrete raw materials indicates properties that are representatives of methods and conditions of combinations and optimisation. Among the many concretes raw materials, superplasticizers, supplementary cementitious materials, alkaline solutions, water-to-binder ratio and the methods of curing influence the behaviour of low carbon concrete. Superplasticizers are useful concrete materials that enhance the workability of concrete. This is because not all water in the concrete during mixing is involved in the hydration process. However, high doses of superplasticizers are known to retard the early ageing of concrete. In this section, the influence of water-to-cement on the optimised model in terms of compressive strength and embodied carbon will be considered. 

### 3.5. Effect of Water on Embodied Carbon and Compressive Strength

Most of the water used in concrete does not take part in the hydration process due to the chemical composition of the raw materials and the sulphate content of the water. The water used for concrete is often described as free water due to the lack of undesirable inorganic or organic substances such as sulphates, etc. The formation of cement paste from the hydration process of concrete formation is dependent on the setting time because of the quality of the water which eventually determines the concrete strength development pattern. The weather condition effects on the concrete strength development in extremely cold weather where the use of heated water may be necessary, have been studied by Brocklesby and Davison [77]. Water as an essential for cement hydration is highly controlled in the concrete formation process and for normal concrete, it is agreed as shown from previous studies that decreasing water in concrete ultimately increases strength, hence the production of concrete from industries is based on minimising water consumption and making the best use of the sustainable water resources. However, the consciousness of associated embodied carbon in concrete and its propagation with increased water content will define the trajectory of new phases of low carbon concrete. While complimenting the efforts of various studies [76,78,79,80,81] on the effect of water on normal concrete where the consideration for associated embodied carbon was either minimal or negligible, there is a need to establish the desired pattern for water in embodied carbon and concrete strength to correlate future results. This section presents the variability of water from selected literature on embodied carbon and concrete strength. From Figure 10, it is shown that concrete strength of about 47 MPa and embodied carbon of about 210 kg-co_2e_/kgm^3^ can be achieved with a water content of 95 kg. it further shows that for an increase in compressive strength, there is an increase in the water content for LCC with a similar increase in the embodied carbon. The optimal water content of 95 kg stabilizes the point where embodied carbon can be reduced to enhance the structural integrity of the mix.

The same observation is shown in Figure 11, Figure 12, Figure 13 and Figure 14 with an optimal water content of 80 kg, 95 kg, 80 kg and 60 kg, respectively.

Concrete shrinkage increases with higher water content for normal concrete; however, for low carbon concrete where the concrete bleeding occurs because of secondary cementitious materials, it is likely to be more and the remaining water that is not consumed by the hydration process will contribute to drying shrinkage. The embodied carbon contained in a concrete mix is dependent on the water content in the mix. As shown from previous studies, concrete embodied carbon, and the physical properties of binders, as shown in Table 18, determine the mechanical response of concrete. The variability of water on embodied carbon and concrete strength as summarised in Table 19 shows the concrete water for the determination of optimal compressive strength and embodied carbon.

### 3.6. Concrete Embodied Carbon Prediction

The physical properties of binder in terms of its surface area and specific gravity define the proportion of fine and coarse particles in it which predict the early age behaviour of concrete. This can be explained by its influence on the formation of capillary porosity, diffusivity, microstructural behaviour, and the extent of water percolation in the concrete. This also controls the hydration reaction as well as the water demand in concrete. Studies have shown that a binder with a higher surface area is obtained from a finer particle size for which the mechanical properties are higher than that of the coarse particle of a smaller surface area [82]. Thus, the use of GGBS as a binder exhibits the characteristic of a binder capable of complete replacement with cement, showing good potential to attain higher structural response in a low carbon concrete regime. Similarly, from the chemical properties of SCM binders in Table 20, a combination of GGBS and silica fume portends an increase in SiO_2_, and Al_2_O_3_ for which the influence on the microstructural enhancement of concrete will result in an increase in the mechanical behaviour of low carbon concrete.

Following the dependence of embodied carbon and concrete strength on the water-to-binder ratio, it was shown in Equation (A1). See Appendix A for detailed derivation. 

In the above equation, the function $\int^{\;}\left(x\right)$ is the embodied carbon and x is the water-to-cement ratio, where B is the constant of proportionality taken as −6.2. The extent of prediction in comparison to another model is shown in Table 21 which shows a good correlation with other models. An assessment of potential embodied carbon in a concrete mix will act as a control in mitigating and controlling the carbon footprint of concrete at the design stage.

## 4. Result and Discussion 

It is observed that there is an increase in the compressive strength from 32.25 MPa to 51.21 MPa when the GGBS to Cement ratio increases from 1.0 to 1.25. This can be explained by the effect of fine GGBS particles on the hydration index. Dai et al. [84] concluded that an increase in the specific area of GGBS decreases the content of Ca(OH)_2_ in the paste, followed by a decline in the compressive strength. When GGBS is used in combination with other binders, such as cement, there is a reduction in the fineness of the composite binder which aids the hydration process on the nanoscale. The increase in water demand for the lightweight aggregate concrete is responsible for the decline in compressive strength as it enhances the increase in the specific area of the binder which allows for porosity that eventually causes high embodied carbon and low compressive strength. However, this effect is mitigated when used in combination with silica fume with its high pozzolanic potential which tends to increase the hydration process and eventually lead to early age strength. It was also noted that for an increase in silica fume from 22.5 to 150 kg/m^3^, there was an increase in compressive strength from 30.292 MPa to 50.33 Mpa; however, embodied carbon increases to 249.04 kg-co2e/kgm^3^ from151.38 kg-co2e/kgm^3^.

Concrete mix design using alkali-activated concrete (AAC) and Portland Cement Concrete (PCC), using a combination of metakaolin, Sodium hydroxide, silica fume and sodium silicate shows a better structural performance. When the cement was completely replaced with 100% of metakaolin at 500 kg/m^3^ and water of 200 kg/m^3^, an embodied carbon of 315 kg-Co_2e_/kgm^3^ was observed with a concrete of 36.737 MPa. Sodium hydroxide and silica fume impact on the embodied carbon was minimal; however, a concrete strength increase was noticed. Further, an increase in cement and a reduction in the metakaolin resulted in an increase in embodied carbon to 409.35 kg-Co_2e_/kgm^3^ at a concrete strength of 63.45 MPa. This was also in combination with 100 kg/m^3^ of silica fume. On increasing the silica fume to 300 kg/m^3^ at no cement added but 100 kg/m^3^ of Metakaolin, the concrete strength increases to 86.97 MPa at an embodied carbon of 267.16 kg-Co_2e_/kgm^3^.

Consideration of the impact of water on concrete is necessary for the evaluation of the extent of embodied carbon from the concrete mix. 

The hydration process aided by the penetration of water in the concrete causes the leaching of calcium hydroxide from the concrete due to its high solubility [75]. The rate of leaching of calcium determines the degradation damage as well as the enlargement of the concrete porosity causing a distortion in the matrix which eventually reduces compressive strength [85]. During the curing process at second hydration, the formed calcium hydroxide reacted with the water during curing to form calcium silicate hydrate (C–S–H) and calcium–hydrate–aluminate (C–H–A) gel, which increases the compactness of the slag concrete and enhances its compressive strength. Concrete compressive strength with 20% fly ash was found to be higher than that of the concrete without admixtures and at 30% it shows reduced strength [86]. The mechanical behaviour of Alkali Activated Concrete (AAC) as compared with that of Portland cement concrete (PCC) exhibited a reduction in embodied carbon to about 30% while a compressive strength of about 100 MPa is observed. It was shown from the optimisation process, that unlike the PCC, where a reduction in water-to-cement reduces embodied carbon, for AAC, the required water absorption of concrete to attain optimality is different from that of the PCC. This is an indication that for pozzolanic activities to increased in SCM, an increase in the water content will facilitate the attainment of the requisite amount of heat of hydration necessary for effective binding. GGBS and Metakaolin were used interchangeably and about 40% of sodium silicate solid was consistent in all the mixes. It further demonstrates that AAC is a potential replacement for PCC which shows high capability in reducing embodied carbon and improving concrete strength. The strength of the concrete matrix is enhanced with the formation of calcium–silicate–hydrate (C–S–H) and calcium–hydrate–Aluminate (C–H–A) gel [87].

### 4.1. Concrete Strength Embodied Carbon Behaviour

The impact of water content on the concrete mix depends on the nature of the component mix constituent. The lightweight aggregate has a high water-absorbing tendency due to the use of recycled or supplementary aggregate. It also portends to the high variability of water content in determining its compressive strength. For normal concrete with a density of about 2400 kg/m^3^, there is decreasing compressive strength with an increasing water cement-to-cement ratio; however, for lightweight aggregate concrete, depending on the water absorption of the aggregate, there may be a need to increase the water content. This makes it difficult to conclude whether increasing or decreasing the water–cement ratio increases the concrete compressive strength for a low carbon concrete. The correlation of the function in Equation (A1) which is derived based on the low compressive strength to high water–cement ratio, attests to this variability for low carbon concrete and predicts the influence of water on concrete embodied carbon.

It is observed from the mix optimisation, that the combination of GGBS and silica fume result in a concrete mix that has reduced embodied carbon at optimal compressive strength. Similarly, it is also shown that the use of alkaline activated concrete (AAC) or the inclusion of an alkaline solution influences the production of alumina-silicate gel, which is responsible for the enhancement of the compressive strength of low carbon concrete. Regarding the influence of water in the concrete mix, optimal water in the concrete of 83 kg/m^3^ is required to attain the minimum embodied carbon of 182.92 kg-CO_2e_/m^3^ at a compressive strength of 39.73 Mpa; however, the water absorption for a low carbon concrete is highly determined by the chemical and physical properties of the SCM binder material. 

### 4.2. Limitation of the Study

There are some other constraints that determine the behaviour of concrete, that were not considered in the linear algorithm as presented in this study, which can affect the prediction of the behaviour of concrete for small datasets. 

Further studies should consider the ability of how concrete behaviour can positively respond to changes in humidity, ambient temperature, and weather condition to reduce cost and carbon emission while considering a small dataset and concrete binders, determined accordingly.

## 5. Conclusions 

The optimisation of low carbon concrete was proposed in this study using the simplex algorithm. A set of concrete raw materials were evaluated and used as input variables which significantly impact the compressive strength and embodied carbon as the outputs. A MATLAB script was developed to solve the algorithm and for each iteration, the inputs were varied at a target and desired output. Different mix proportions were tried, and the corresponding outputs were measured. Based on the results of this study, the following conclusions were obtained.

The water-to-binder ratio is a significant parameter that determines the embodied carbon of low carbon concrete. The determination of the embodied carbon value of low carbon concrete at the trial mix batch will minimise the cement overdose of the concrete and reduce the assumed uncertainties inherent in over-design.The water absorption rate for LCC is higher than the normal weight concrete for optimal compressive strength to be attained at a low carbon value,Alkaline activated concrete (AAC) offers the potential for a complete replacement of cement for sustainable, eco-friendly, low carbon concrete with good mechanical performance.The use of an alkaline solution enhances the hydration process of GGBS which is evident with the increase in compressive strength when used in combinationThe combination of two alternative binders for LCC improves the particle size fineness and eventually improves the binder hydration process which is optimal at the nanoscale.The combination of GGBS and silica fume portends the chance of a complete replacement of cement as a concrete binderIt is observed that GGBS has a comparative advantage over fly ash in terms of carbon reduction in concrete.

## Figures and Tables

**Figure 1 materials-15-08673-f001:**
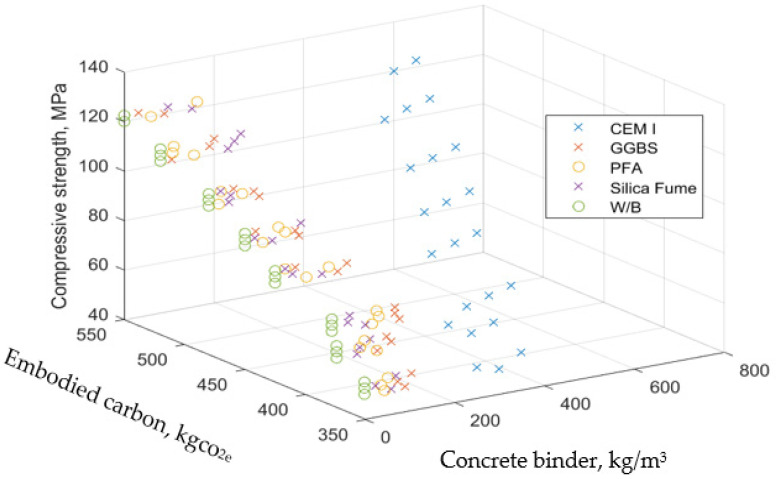
influence of input parameters on compressive strength and embodied carbon.

**Figure 2 materials-15-08673-f002:**
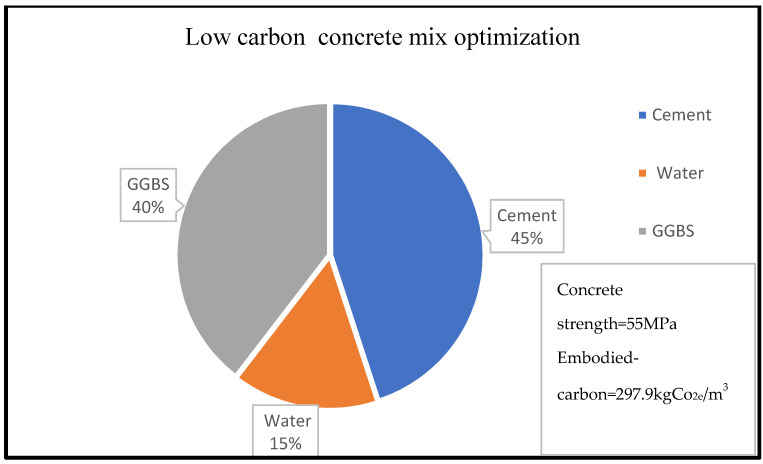
LCC constituent material at water content of 15% [34].

**Figure 3 materials-15-08673-f003:**
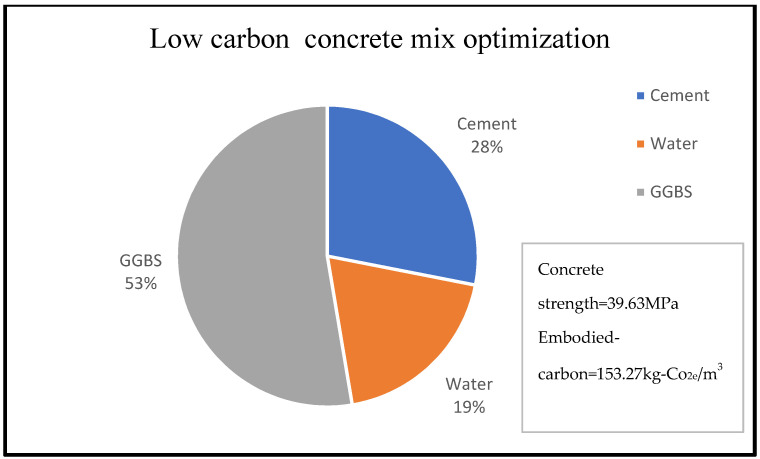
LCC constituent material at water content of 19% [33].

**Figure 4 materials-15-08673-f004:**
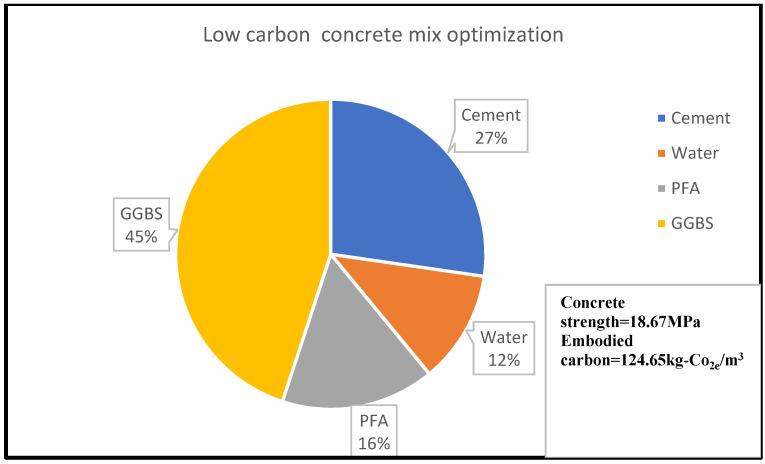
LCC constituent material at water content of 16% [22].

**Figure 5 materials-15-08673-f005:**
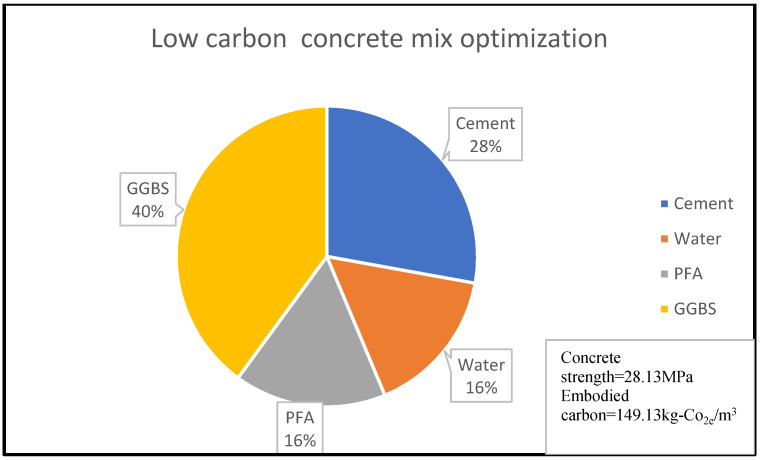
LCC constituent material at water content of 16% for Concrete Batch A [32].

**Figure 6 materials-15-08673-f006:**
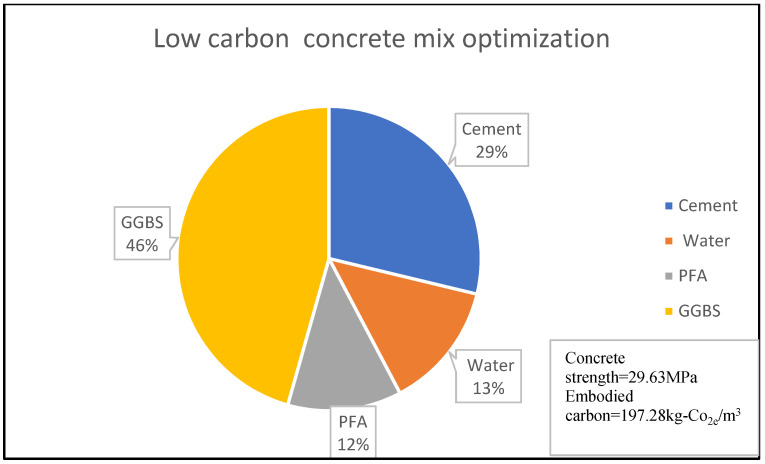
LCC constituent material at water content of 13% for Concrete Batch B [32].

**Figure 7 materials-15-08673-f007:**
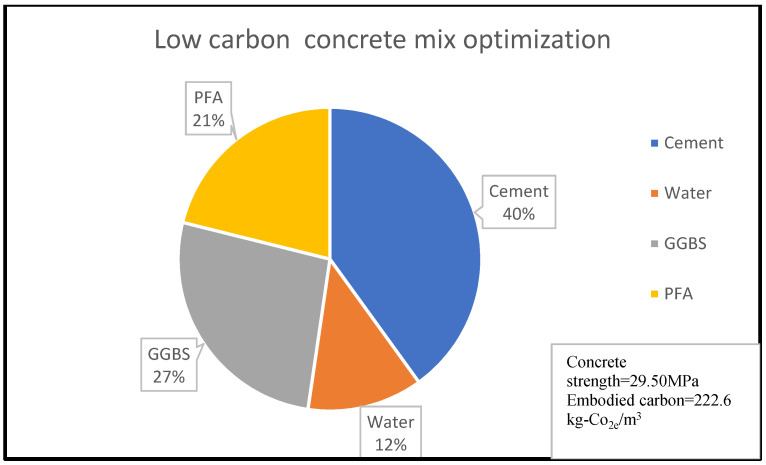
LCC constituent material at water content of 12% for Concrete Batch C [32].

**Figure 8 materials-15-08673-f008:**
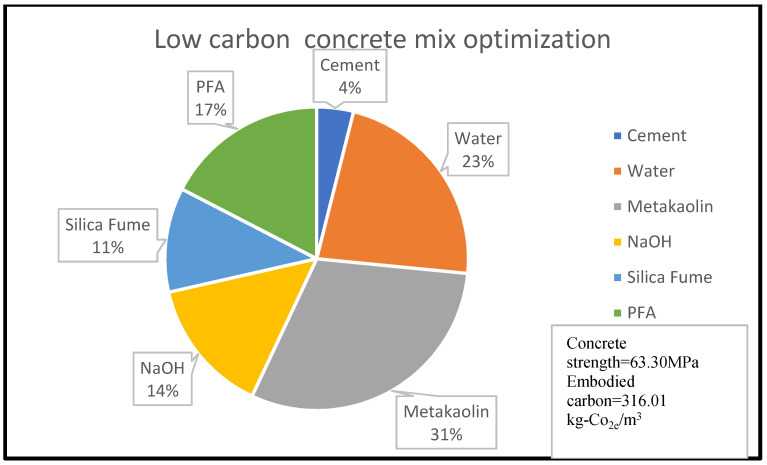
AAC constituent material at water content of 23% [35].

**Figure 9 materials-15-08673-f009:**
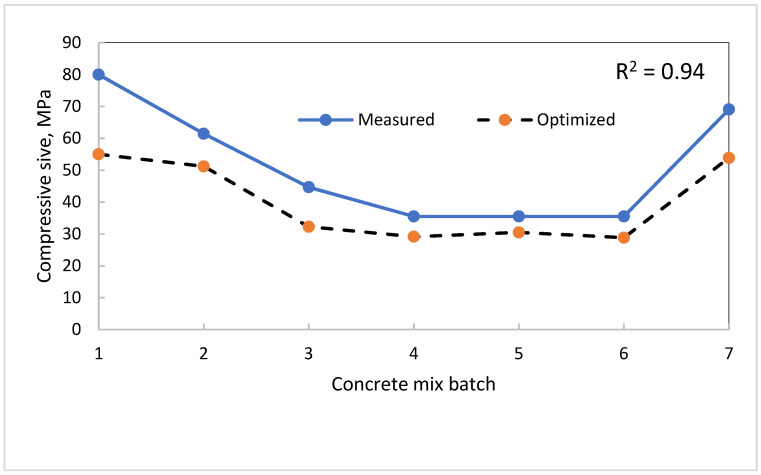
Optimised Embodied Carbon relation with measured compressive strength.

**Figure 10 materials-15-08673-f010:**
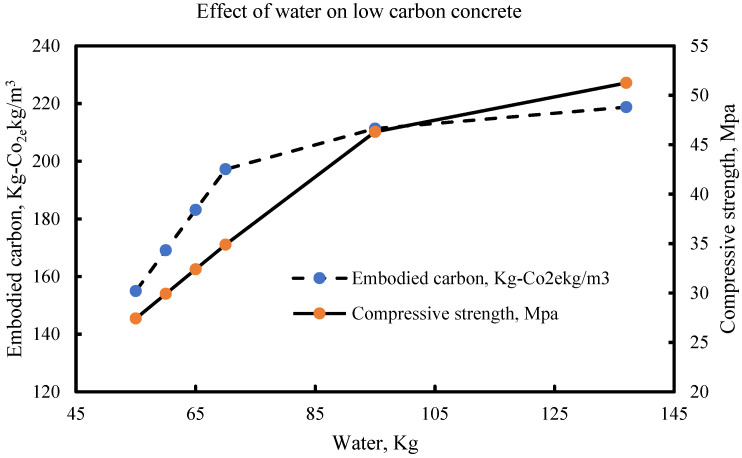
LCC at optimal water content of 85-105 litres per m^3^ of concrete [33].

**Figure 11 materials-15-08673-f011:**
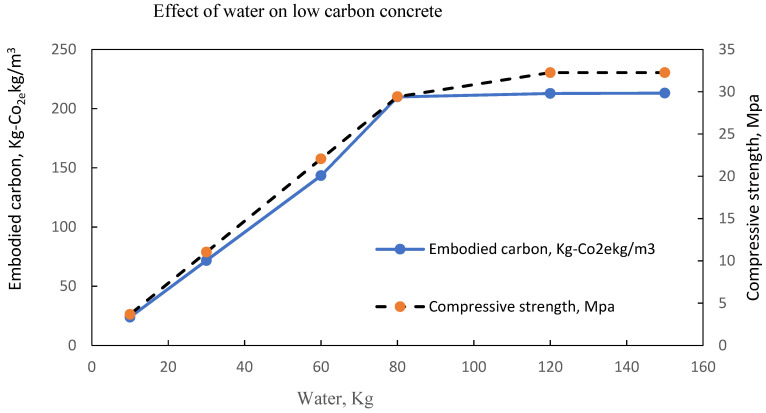
LCC at optimal water content of 80 litres per m^3^ of concrete [22].

**Figure 12 materials-15-08673-f012:**
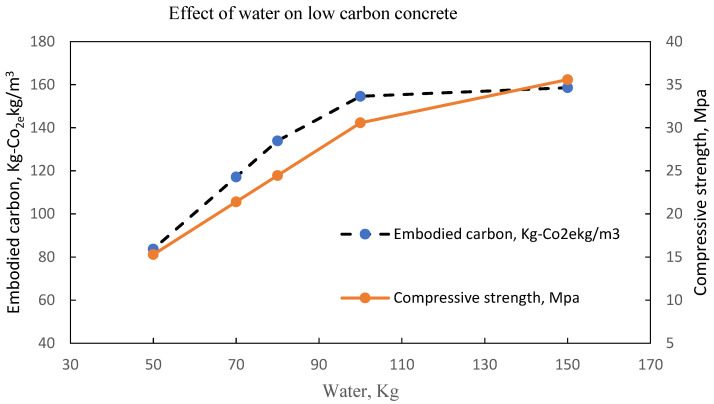
LCC at optimal water content of 130-150 litres per m^3^ of concrete Batch A [32].

**Figure 13 materials-15-08673-f013:**
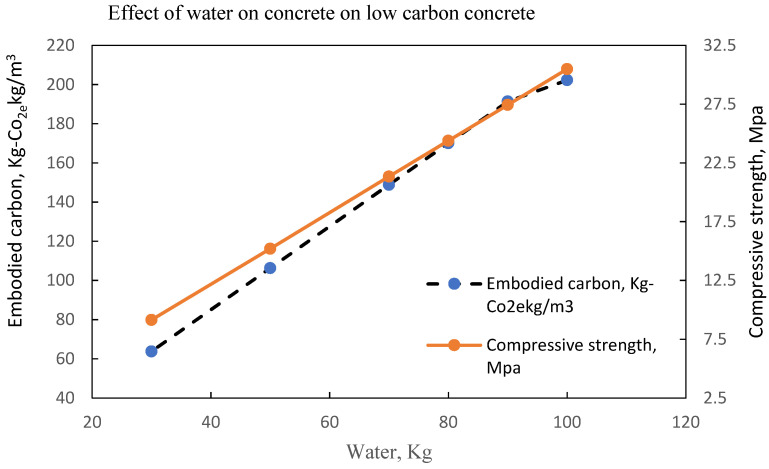
LCC at optimal water content of 80-100 litres per m^3^ of concrete Batch B [32].

**Figure 14 materials-15-08673-f014:**
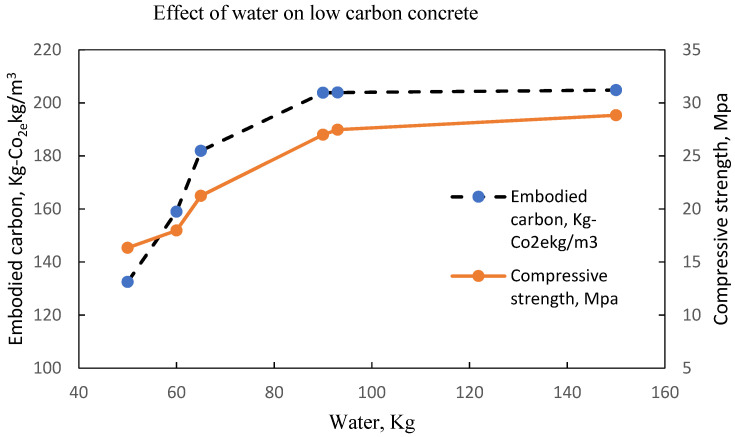
LCC at optimal water content of 80-95 litres per m^3^ of concrete Batch C [32].

**Table 1 materials-15-08673-t001:** Density and thermal conductivity values of some natural fibres used to produce LWC [40].

Fibre Raw Materials	Bulk Density (Kg/m)	Thermal Conductivity λ (w/mK)
Cork	120–180	0.085
Corn Cob Panel		0.139
Sugarcane	100–125	0.0469–0.0496
Cellulose Loose fill		0.05
Flax	5–50	0.038–0.075
Cellulose (Recycled paper)	30	0.041
Hemp	20–45	0.040–0.060
Straw bales	102.6	0.067
Coconut	85	0.085

**Table 2 materials-15-08673-t002:** Embodied coefficient for LCC Materials [45].

Materials		Embodied Carbon, (Kg CO_2_/tonne)
Portland cement, CEM1		860
Secondary cementitious Materials	Ground granular base slag (GGBS)	79.6
	Fly ash	0.1
	Limestone	8
Aggregate		2.6

**Table 3 materials-15-08673-t003:** Effect of Cement type on ECO_2_ content of designated concretes [45].

Concrete	Concrete Type (Slump Class)	ECO_2_ (kg CO_2_/m^3^)		
		CEMI Concrete	30% PFA Concrete	50% GGBS Concrete
Blinding, mass fill, strip footings, mass foundations	GEN1 (S2)	165	120	95
Reinforced foundations	RC25/30 (S2) **	295	245	190
Ground floors	RC28/35 (S2) *	295	245	175
Structural: in-situ, superstructure, walls, basements	RC32/40 (S2) *	345	295	220
Higher strength concrete	RC 40/50 (S2) **	405	330	255

* Includes 30 kg/m^3^ steel reinforcement, ** includes 100 kg/m^3^ steel reinforcement.

**Table 4 materials-15-08673-t004:** LCC Rating [58].

EXAMPLE OF AN EMBODIED CARBON RATING CERTIFICATE
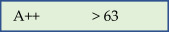	Cement type	IIA
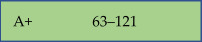 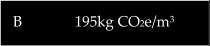	SCM	GGBS
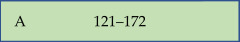	Cement content	300 kg/m^3^
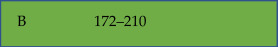	W/C	0.65
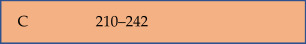	SCM Content	40%
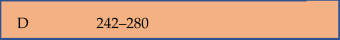	Aggregate size	20 mm
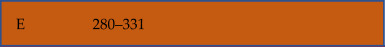	Admixtures	Superplasticizer
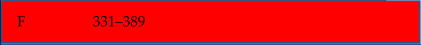	Slump class	S4
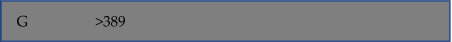	**Strength** **C28/35**	

**Table 5 materials-15-08673-t005:** Distribution of embodied carbon fractiles (kg CO_2_ e/m^3^) [63].

Rating	Kg Co_2_e/m^3^ Fractiles Range within the Strength Class
A^++^	Kg Co_2_e/m^3^ below those of benchmarked concrete
A^+^	0–5%
A	5–20%
B	20–40%
C	40–60%
D	60–80%
E	80–95%
F	95–100%
G	Kg Co_2_e/m^3^ above those of benchmarked concrete

**Table 6 materials-15-08673-t006:** Typical composition of different cement type [70].

	Ordinary Portland Cement (%)	Portland Fly-Ash Cement (%)	Blast-Furnace Cement (%)	Pozzolanic Cement Mixes (%)
Clinker	95–100	65–94	5–64	45–89
Fly-ash (waste residues from coal-fired power stations)		6–35		
Blast furnace slag			36–95	
Pozzolana (volcanic ash)				11–55
Other constituents (e.g., clinker dust, other mineral additives)	0–5	0–5	0–5	0–5

**Table 7 materials-15-08673-t007:** Embodied Carbon for Concrete mix.

Concrete Component	Embodied Carbon Value (kgCo_2e_/kg)	Source
Port Land Cement	0.93	[71]
Sand	0.0048	[71]
Crushed aggregate	0.0012	[71]
Superplasticizer	2.388	[34]
Ground granular base aggregate (GGBS)	0.135	[34]
Fly ash	0.020	[71]

**Table 8 materials-15-08673-t008:** Limit of the input variables used.

Input Variables	Limit
Ordinary Portland cement (OPC)	0–650 kg/m^3^ with an increment of 25
Fine aggregate (FA)	260–969 kg/m^3^ with an increment of 50
Coarse Aggregate (CA)	679–1400 kg/m^3^ with an increment of 50
Water (W)	0–195 kg/m^3^ with an increment of 50
Chemical admixture (CH.A)	0–45 kg/m^3^ with an increment of 5
Ground granular base slag (GGBS)	0–320 kg/m^3^ with an increment of 50
Pulverised fly ash (PFA)	0–474 kg/m^3^ with an increment of 100
Sodium hydroxide (NAOH)	0–616 kg/m^3^ with an increment of 25
Metakaolin (MEK)	0–289 kg/m^3^ with an increment of 100
Sodium silicate (NaSi)	0–239 kg/m^3^ with an increment of 50
Silica fume (SF)	0–180 kg/m^3^ with an increment of 25
Water-to-cement ratio (W/C)	0–0.87 with an increment of 0.01

**Table 9 materials-15-08673-t009:** Concrete mix optimisation extracted from [34].

	Concrete Mix (Kg/m^3^)													
	1	2	3	4	5	6	7	8	9	10	11	12	
OPC	300	300	300	350	400	450	500	600	300	350	400	450	
FA	650	650	650	650	650	650	650	650	650	650	650	650	
CA	1400	1400	1400	1400	1400	1400	1400	1400	1400	1400	1400	1400	
W	45	45	45	45	45	45	45	45	45	45	45	45	
CH.A	45	45	45	45	45	45	45	45	45	45	45	45	
GGBS	90	90	150	140	120	112.5	25	90	75	122.5	140	160	
PFA	45	90	90	70	40	22.5	75	60	37.5	90	105	120	
NAOH	0	0	0	0	0	0	0	0	0	0	0	0	
MEK	0	0	0	0	0	0	0	0	0	0	0	0	
NaSi	0	0	0	0	0	0	0	0	0	0	0	0	
SF	60	45	75	105	60	45	150	150	25	52.5	35	40	
W/C	0.15	0.18	0.13	0.15	0.11	0.08	0.10	0.09	0.08	0.08	0.07	0.08	
EC	350	370	400	425	450	480	520	550	300	350	375	420	
CS	50	60	70	80	90	100	110	120	55	65	75	85	
Optimal Mix(Ec = 297.924 kg-co_2e_/kgm^3^, CS = 55 MPa													
OPC	FA	CA	W	CH.A	GGBS	PFA	NAOH	MEK	NaSi	SF	W/C	OPCS	EC
275	650	1400	200	45	25	27.5	0	0	0	150		30.292	151.38

OPC = Ordinary Portland Cement, FA = Fine aggregate, CA = Coarse aggregate, w = Water, CH.A = Chemical admixture, GGBS = Ground granular base slag, PFA = Pulverised fly ash, NAOH = Sodium hydroxide solid, MEK = Metakaolin, NaSi = Sodium silicate, SF = Silica fume, EC = Embodied carbon, CS = Concrete compressive strength.

**Table 10 materials-15-08673-t010:** Concrete mix optimisation extracted from [33].

	Concrete Mix (Kg/m^3^)											
	1	2	3	4	5	6	7	8	9	10	11	12
OPC	360	108	72	72	72	180	120	80	120	80	136	136
FA	350	350	350	350	350	350	350	350	350	350	350	350
CA	720	720	720	720	720	720	720	720	720	720	720	720
W	137	137	137	137	137	137	137	137	137	137	137	137
CH.A	8.8	8.2	6.8	9.2	6	5.2	5.2	5.6	5	5.2	7.6	7.5
GGBS	0	252	288	288	288	180	280	320	280	320	204	204
PFA	0	0	0	0	0	10.8	0	0	0	0	0	0
NAOH	0	0	0	0	0	0	0	0	0	0	0	0
MEK	0	0	0	0	0	0	0	0	0	0	0	0
NaSi	0	0	0	0	0	0	0	0	0	0	0	0
SF	0	0	0	0	0	0	0	0	0	0	0	0
W/C	0.38	0.54	0.47	0.47	0.47	0.76	0.49	0.43	0.49	0.43	0.67	0.67
EC	386	183	153	154	153	147	196	164	209	176	202	184
CS	74.3	68	66	68.3	54	51.3	60.8	56.3	55.8	49.5	67.5	65.5
Optimal Mix(Ec = 218.76 kg-co_2e_/kgm^3^, CS = 51.21 MPa												
OPC	FA	CA	W	CH.A		GGBS		PFA	NAOH	MEK	NaSi	SF
200	350	720	137	5		250		0	0	0	0	0

OPC = Ordinary Portland Cement, FA = Fine aggregate, CA = Coarse aggregate, w = Water, CH.A = Chemical admixture, GGBS = Ground granular base slag, PFA = Pulverised fly ash, NAOH = Sodium hydroxide solid, MEK = Metakaolin, NaSi = Sodium silicate, SF = Silica fume, EC = Embodied carbon, Concrete compressive strength.

**Table 11 materials-15-08673-t011:** Concrete mix optimisation extracted from [22].

	Concrete Mix (Kg/m^3^)											
	1	2	3	4	5	6	7	8	9	10	11	12
OPC	205	175	169	148	155	147	205	175	169	148	155	147
FA	809	869	870	846	825	828	809	869	870	846	825	828
CA	588	633	655	640	675	698	588	633	655	640	675	698
W	200	168	157	164	150	136	200	168	157	164	150	136
CH.A	1.65	3.29	4.74	4.23	4.51	5.4	1.65	3.29	4.74	4.23	4.51	5.4
GGBS	205	175	169	222	155	147	205	175	169	222	155	147
PFA	0	0	0	0	0	0	0	0	0	0	0	0
NAOH	0	0	0	0	0	0	0	0	0	0	0	0
MEK	0	0	0	0	0	0	0	0	0	0	0	0
NaSi	0	0	0	0	0	0	0	0	0	0	0	0
SF	0	0	0	0	0	0	0	0	0	0	0	0
W/C	0.98	0.96	0.93	1.11	0.97	0.93	0.98	0.96	0.93	1.11	0.97	0.93
EC	348	326	326	307	325	325	348	326	326	307	325	325
CS	42.5	37.5	46	46	46	50	42.5	37.5	46	46	46	50
Optimal Mix (Ec = 213.06 kg-co_2e_/kgm^3^, CS = 32.258 MPa)												
OPC	FA	CA	W	CH.A		GGBS		PFA	NAOH	MEK	NaSi	SF
200	809	588	150	5		200		0	0	0	0	0

OPC = Ordinary Portland Cement, FA = Fine aggregate, CA = Coarse aggregate, w = Water, CH.A = Chemical admixture, GGBS = Ground granular base slag, PFA = Pulverised fly ash, NAOH = Sodium hydroxide solid, MEK = Metakaolin, NaSi = Sodium silicate, SF = Silica fume, EC = Embodied carbon, Concrete compressive strength.

**Table 12 materials-15-08673-t012:** Concrete mix optimisation extracted from Concrete Batch A [32].

	Concrete Mix (Kg/m^3^)											
	1	2	3	4	5	6	7	8	9	10	11	12
OPC	236	200	249	181	196	219	258	216	271	197	213	246
FA	909	930	959	931	910	903	916	925	969	932	884	860
CA	958	933	938	962	993	997	887	885	864	895	965	950
W	161	167	151	158	157	157	177	180	169	171	170	178
CH.A	1.22	1.9	1.97	1.3	1.83	1.81	1.33	2.04	2.16	1.41	1.98	1.59
GGBS	0	33	0	39	39	0	0	35	0	42	42	0
PFA	35	38	34	39	26	39	39	41	37	42	28	43
NAOH	0	0	0	0	0	0	0	0	0	0	0	0
MEK	0	0	0	0	0	0	0	0	0	0	0	0
NaSi	0	0	0	0	0	0	0	0	0	0	0	0
SF	0	0	0	0	0	0	0	0	0	0	0	0
W/C	0.68	0.84	0.61	0.87	0.80	0.72	0.69	0.83	0.62	0.87	0.80	0.72
EC	243.24	211.38	252.76	192.57	206.54	227.25	265.81	228.04	277.32	209.24	224.19	254.82
CS	18	24	30	45	48	48	18	24	30	45	48	48
Optimal Mix (Ec = 154.59 kg-co_2e_/kgm^3^, CS = 29.13 MPa												
OPC	FA	CA		W	CH.A	GGBS	PFA	NAOH		MEK	NaSi	SF
150	860	864		100	1.22	35	40	0		0	0	0

OPC = Ordinary Portland Cement, FA = Fine aggregate, CA = Coarse aggregate, w = Water, CH.A = Chemical admixture, GGBS = Ground granular base slag, PFA = Pulverised fly ash, NAOH = Sodium hydroxide solid, MEK = Metakaolin, NaSi = Sodium silicate, SF = Silica fume, EC = Embodied carbon, Concrete compressive strength.

**Table 13 materials-15-08673-t013:** Concrete mix optimisation extracted from Concrete Batch B [32].

	Concrete Mix (Kg/m^3^)											
	1	2	3	4	5	6	7	8	9	10	11	12
OPC	268	266	264	284	264	272	334	296	334	260	282	324
FA	872	835	896	896	896	902	787	808	884	832	792	810
CA	932	925	931	931	940	957	954	933	915	954	997	969
W	160	176	160	160	160	160	171	170	158	160	161	164
CH.A	3.05	2.92	2.64	2.68	2.64	2.72	2.95	3.2	2.66	1.86	2.64	2.67
GGBS	34	55	33	34	33	34	0	48	0	56	57	0
PFA	34	44	33	17	33	34	59	56	46	56	38	57
NAOH	0	0	0	0	0	0	0	0	0	0	0	0
MEK	0	0	0	0	0	0	0	0	0	0	0	0
NaSi	0	0	0	0	0	0	0	0	0	0	0	0
SF	0	0	0	0	0	0	0	0	0	0	0	0
W/C	0.60	0.66	0.61	0.56	0.61	0.59	0.51	0.57	0.47	0.62	0.57	0.51
EC	275.02	275.88	271.09	289.7	271.1	278.76	337.76	303.57	335.93	268.47	289.29	327.29
CS	18	24	30	45	48	48	18	24	30	45	48	48
	Optimal Mix(Ec = 202.26 kg-Co_2e_/kgm^3^, CS = 30.488 MPa)											
	OPC	FA	CA	W	CH.A	GGBS	PFA	NAOH	MEK	NaSi	SF	
	200	787	915	100	1.22	200	60	0	0	0	0	

OPC = Ordinary Portland Cement, FA = Fine aggregate, CA = Coarse aggregate, w = Water, CH.A = Chemical admixture, GGBS = Ground granular base slag, PFA = Pulverised fly ash, NAOH = Sodium hydroxide solid, MEK = Metakaolin, NaSi = Sodium silicate, SF = Silica fume, EC = Embodied carbon, Concrete compressive strength.

**Table 14 materials-15-08673-t014:** Concrete mix optimisation extracted from Concrete Batch C [32].

	Concrete Mix(Kg/m^3^)											
	1	2	3	4	5	6	7	8	9	10	11	12
OPC	349	334	268	293	333	383	387	387	408	408	408	405
FA	885	884	833	781	800	679	814	785	770	786	767	924
CA	880	914	929	981	957	969	929	874	911	918	871	901
W	165	158	165	167	169	187	160	160	160	160	160	153
CH.A	2.78	2.66	1.92	2.74	2.74	3.16	4.83	6.31	6.12	5.1	6.63	4.37
GGBS	0	0	58	59	0	0	49	49	51	51	51	65
PFA	46	46	58	39	59	68	49	49	51	51	51	65
NAOH	0	0	0	0	0	0	0	0	0	0	0	0
MEK	0	0	0	0	0	0	0	0	0	0	0	0
NaSi	0	0	0	0	0	0	0	0	0	0	0	0
SF	0	0	0	0	0	0	0	0	0	0	0	0
W/C	0.47	0.47	0.62	0.57	0.51	0.49	0.41	0.41	0.39	0.39	0.39	0.38
EC	350.89	335.93	276.68	300.44	336.54	386.04	388.66	388.3	408.89	408.65	408.96	405.39
CS	18	24	30	45	48	48	18	24	30	45	48	48
	Optimal Mix(Ec = 204.81 kg-CO_2e_/kgm^3^, CS = 28.829 MPa)											
OPC	FA	CA	W	CH.A		GGBS	PFA		NAOH	MEK	NaSi	SF
200	700	900	150	1.92		200	60		0	0	0	0

OPC = Ordinary Portland Cement, FA = Fine aggregate, CA = Coarse aggregate, w = Water, CH.A = Chemical admixture, GGBS = Ground granular base slag, PFA = Pulverised fly ash, NAOH = Sodium hydroxide solid, MEK = Metakaolin, NaSi = Sodium silicate, SF = Silica fume, EC = Embodied carbon, Concrete compressive strength.

**Table 15 materials-15-08673-t015:** Concrete mix optimisation extracted from [35].

	Concrete Mix (Kg/m^3^)											
	1	2	3	4	5	6	7	8	9	10	11	12
OPC	0	460	0	505	0	228	0	460	0	505	0	228
FA	651	622	796.3	630	793	800	651	622	796.3	630	793	800
CA	1209	1105	1055.3	1030	793	1110	1209	1105	1055.3	1030	793	1110
W	0	193	3.9	195	163	131	0	193	3.9	195	163	131
CH.A	0	0	0	0	0	0	0	0	0	0	0	0
GGBS	60	0	0	0	0	0	60	0	0	0	0	0
PFA	340	0	0	60	474	0	340	0	0	60	474	0
NAOH	45.7	0	24.2	1.3	61.6	0	45.7	0	24.2	1.3	61.6	0
MEK	0	0	289	0	0	0	0	0	289	0	0	0
NaSi	114	0	238.9	0	0	0	114	0	238.9	0	0	0
SF	0	0	0	0	46.2	45	0	0	0	0	46.2	45
W/C	0.00	0.42	0.00	0.39	0.00	0.57	0.00	0.42	0.00	0.39	0.00	0.57
EC	395	222	435	512	247	639	395	222	435	512	247	639
CS	40	40	58.5	65	106	105	40	40	58.5	65	106	105
	Optimal Mix(Ec = 369.56 kg-Co_2e_/kgm^3^, CS = 53.88 MPa											
OPC	FA	CA	W	CH.A		GGBS	PFA	NAOH		MEK	NaSi	SF
50	650	800	100	0			0	0	80	500	150	60

OPC = Ordinary Portland Cement, FA = Fine aggregate, CA = Coarse aggregate, w = Water, CH.A = Chemical admixture, GGBS = Ground granular base slag, PFA = Pulverised fly ash, NAOH = Sodium hydroxide solid, MEK = Metakaolin, NaSi = Sodium silicate, SF = Silica fume, EC = Embodied carbon, Concrete compressive strength.

**Table 16 materials-15-08673-t016:** Summary of the optimised Embodied Carbon using the LP Program.

Embodied Carbon(Kg-Co_2e_/kg-m^3^)	Optimised EmbodiedKg-Co_2e_/kg-m^3^)	Reduction in Embodied Carbon
434.8333	297.92	136.91
192.25	218.76	-26.51
326.1667	213.06	113.1067
232.7633	154.59	78.17333
293.655	202.26	91.395
366.2808	204.81	161.4708
408.3333	369.56	38.77333
	% Embodied carbon reduction	26%

**Table 17 materials-15-08673-t017:** Summary of the optimised compressive strength using the LP Program.

Concrete Strength, Mpa	Optimised Compressive Strength, Mpa	Loss in Strength Due to Low Carbon
80	55	25
61.44167	51.21	10.23167
44.66667	32.258	12.40867
35.5	29.13	6.37
35.5	30.488	5.012
35.5	28.829	6.671
69.08333	53.88	15.20333
% Reduction in strength		24%

**Table 18 materials-15-08673-t018:** Physical properties of Cement and GGBS [77].

Material	Physical Properties	
	Specific Gravity	Surface Area(cm^2^/g)
CEMI	3.14	3670
GGBS	2.90	4550

**Table 19 materials-15-08673-t019:** Limiting Water in Low carbon concrete.

Figure	Water, Kg	Embodied Carbon, Kg-Co_2e_kg/m^3^	Compressive Strength, Mpa
8	95	211.31	46.3
9	80	225.3	29.412
10	100	154.59	30.575
11	90	191.35	27.439
12	50	132.48	65

**Table 20 materials-15-08673-t020:** Chemical composition of Cement and GGBS [83].

Material	Chemical composition (%)						
	CaO	SiO_2_	Al_2_O_3_	Fe_2_O_3_	MgO	NaO	Others
CEMI	63.8	21.2	5.4	3.2	2.0	0.8	3.6
GGBS	43.8	33.5	9.0	3.6	2.7	0.6	6.8
PFA	2.0	54.0	24.0	8.0	1.3	0.9	9.8
SILICA	0.4	91.5	0.2	0.7	1.5	1.9	3.8

**Table 21 materials-15-08673-t021:** Comparison with other models.

Model		[34]	[33]	[22]
Equation (A1)	R^2^	0.8208	0.8829	0.8842

## Data Availability

Not applicable.

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
