# Peer review of "Optimisation of Embodied Carbon and Compressive Strength in Low Carbon Concrete"

_materials, 2022, doi:10.3390/ma15238673_

Round 1

Reviewer 1 Report

Comments are listed below:

1. Strengthen the abstract section. Add the key conclusion of the works in the last two lines of the abstract section. Remove the unnecessary information.
2. Discuss the novelty of the work in respect of the application  

3. There are numerous spelling and grammatical errors. Please revise the manuscript thoroughly. Sentences are also not complete and references are also cited in a rough manner.  
4. Try to make a bridge between current and previously published work and specify the gap area and objective of the work. The introduction section is very poor: refer to following published work and write this paragraph in introduction section:

"One previous research from Wenchen et al. shows a very good method for determining the experimental parameters. Because structural-level experiments are not like material-level tests, it requires a great amount of manpower and money to support them. Therefore, not only the experimental design (test setup, instruments) due to high experimental requirements but also the experimental parameters are very important for the test. In this paper, all the experimental parameters were chosen based on progressive collapse analysis of the whole building (four-story whole reinforced concrete two-way slab building) with nonlinear finite element analysis other than from other papers."

Ma, Wenchen, Ying Tian, Hailong Zhao, and Sarah L. Orton. "Time-Dependent Behavior of Reinforced Concrete Columns Subjected to High Sustained Loads." Journal of Structural Engineering 148, no. 10 (2022): 04022161.

5. Provide the image of the experimental setup with good quality. Also, add the image of the welded pipe produced.

6. The results are ok but the discussion section is very poor. It looks like a technical report instead of a technical article. Improve the discussion section and add more references in support of the results.

7. Shorten the length of the conclusion section.

8. The work is good, but the technical discussion and introduction section needs improvement. Paper can be accepted after following minor corrections.

Author Response

Kindly see the attached reviewers comments 

Reviewer 2 Report

It is a somewhat interesting topic regarding “Optimization of Embodied Carbon and Compressive Strength in Low Carbon Concrete.” This paper should be revised as per the below comments.

Abstract

·         Expand the results, the results should be better described in the abstract. For example, how and why this study needs to be investigated?

·         Tell the reader what you have done.

·         Explain "how" you have done it.

·         Why is this model preferred over other methods?

Introduction and literature review

·         what are the innovations of this paper, please explain the motivation and the innovation of this paper scientifically.

·         The introduction section needs revisions to be written more coherently and concisely.

·         You should better emphasize the importance of your work.

·         The literature review must be added as a separate section. The literature review must have a logical sequence contemplating the description of the work topic, research problem (threshold of knowledge), justification, hypotheses, and objectives. Also, the papers listed below should be added to the Literature Review section:

https://link.springer.com/chapter/10.1007/978-981-15-0751-9_11

https://www.mdpi.com/1996-1944/15/12/4051

Materials and Methods

 ·         The methods applied in the current paper are unclear; thus, more illustrations must be added.

·         The Methodology section needs more justification and discussion concerning the selected significant variables.

·         The limitations of the methodology and results need substantial improvement for the paper to impact research and practice.

·         Section 3 is missing. Name this section, please.

Results and Discussion

·         How can the authors determine that the MATLAB software is significant to be investigated? And then, what studies have been evaluated to make these claims?

·         The results of this study are not discussed or compared with other studies.

  • How were the related mentioned models used in this paper chosen?

·         The implication of this study is missing.

·         The proposed model should be validated against many cases to be validated

·         The presentation of the results is confusing and not well organized.

·         It is recommended that you add the practical and theoretical implications section and how this model would enhance decision-making by providing examples.

·         The results of your comparative study should be discussed in-depth and with more insightful comments on the behavior of your results on various case studies.

Conclusion

The conclusion of the work should be rephrased and expanded to enhance the readership quality and well written scientifically.

Author Response

Kindly see the attached authors comments 

Reviewer 3 Report

This paper deals with interesting topic on the mechanical behavior of concrete and embodied carbon in low carbon Concrete (LCC). The authors conducted valuable research where an optimization of the embodied carbon and compressive strength was done using simplex algorithm program on MATLAB software. With the results obtained the authors also developed a model to predict the associated embodied carbon of a concrete mix. The results obtained are good, but some information needs to be inserted in the text, and the methodology needs be more detailed to improve the paper. Thus, some suggestions are indicated as follow:

·       The English needs to be improved. There are grammatical errors in the text.

·       Please, verify the number of the sections and subsections. For example, in the text does not exist a section with the number 3.

·       The figures 1 to 7 need to be improved. Adjust the figures text.

·       In table 16 the authors use the term “Savings”, but the concrete strength in LCC is less than in normal concrete. Please, consider using another term.

·       On the subsection “Data Source” is necessary say about the domain of the data, the maximum and minimum of each parameter used in the modeling processes and how the values of embodied carbon were estimated in each literature work. Please, improve your data section.

·       On the section Methodology, you mix the data collected from the literature with the results obtained using linear programming (LP), it would be interesting to separate the results obtained from the methodology.

·       The subsection 3.1 needs to be adjusted.

·       In the subsection 3.1, the authors need explain better what the variable “water” is plotted in the figures 8-12. And the Figures need to be called in the text.

·       In the text, on the line 654, the authors presented the equation 3, but the aren’t the equation 1 and 2 in the text. Please, adjust the equations numbers.

·       Regarding the equation 3, is important describe better the modeling process to obtain the mathematical formulation. Please, explain and show the model process utilized in the modeling. Is important too, describe how the authors obtained the parameter “B=0.62”.

·       The authors said that the table 20 shows a good correlation between the proposed model and other model, but no one model was presented. What model the authors are what model is this that the authors are talking about? Please, explain better your methods and results. Your research is good, but some mistakes in the methodology make that the results questionable. Improve the modeling process.

·       Please, consider revising your conclusions, inserting the main results of your work. You need to write more about your proposed model and about the results obtained in the linear programing (LP). What is new in your research? What made your research different from the others existent in the literature? What is the limitation of your proposed model? What is necessary to generate a model more reliable?

Author Response

(The authors gave the same response as above.)

Round 2

Reviewer 1 Report

Accept 

Author Response

There is no reviewer 1 comment

Reviewer 2 Report

The introduction section needs revisions to be written more coherently and concisely.

The authors should better emphasize the importance of their work.

The literature review must be added as a separate section. The literature review must have a logical sequence contemplating the description of the work topic, research problem (threshold of knowledge), justification, hypotheses, and objectives. Also, the papers listed below should be added to the Literature Review section:

https://www.sciencedirect.com/science/article/abs/pii/S1474034618306475

https://link.springer.com/chapter/10.1007/978-981-15-0751-9_11

https://www.mdpi.com/1996-1944/15/12/4051

https://www.researchgate.net/profile/Victor-Yepes/publication/361820131_CO2-Optimization_of_Post-Tensioned_Concrete_Slab-Bridge_Decks_Using_Surrogate_Modeling/links/62c6f71ed7bd92231f9e4f52/CO2-Optimization-of-Post-Tensioned-Concrete-Slab-Bridge-Decks-Using-Surrogate-Modeling.pdf

https://www.sciencedirect.com/science/article/abs/pii/S0959652617314154

https://www.mdpi.com/2075-5309/12/8/1166

Author Response

Find attached authors response for reviewer 2

Reviewer 3 Report

The authors made all the suggestions/corrections that were requested. The paper can be accept in present form.

Author Response

There is no reviewer 3 comment